# Good Cholesterol Gone Bad? HDL and COVID-19

**DOI:** 10.3390/ijms221910182

**Published:** 2021-09-22

**Authors:** George E. G. Kluck, Jeong-Ah Yoo, Emmanuel H. Sakarya, Bernardo L. Trigatti

**Affiliations:** Thrombosis and Atherosclerosis Research Institute and Department of Biochemistry and Biomedical Sciences, McMaster University and Hamilton Health Sciences, Hamilton, ON L8L 2X2, Canada; kluckgeg@gmail.com (G.E.G.K.); yoojeongah92@gmail.com (J.-A.Y.); sakaryae@mcmaster.ca (E.H.S.)

**Keywords:** SARS-CoV-2, COVID-19, HDL, SR-B1

## Abstract

The transmissible respiratory disease COVID-19, caused by the severe acute respiratory syndrome coronavirus 2 (SARS-CoV-2), has affected millions of people worldwide since its first reported outbreak in December of 2019 in Wuhan, China. Since then, multiple studies have shown an inverse correlation between the levels of high-density lipoprotein (HDL) particles and the severity of COVID-19, with low HDL levels being associated with an increased risk of severe outcomes. Some studies revealed that HDL binds to SARS-CoV-2 particles via the virus’s spike protein and, under certain conditions, such as low HDL particle concentrations, it facilitates SARS-CoV-2 binding to angiotensin-converting enzyme 2 (ACE2) and infection of host cells. Other studies, however, reported that HDL suppressed SARS-CoV-2 infection. In both cases, the ability of HDL to enhance or suppress virus infection appears to be dependent on the expression of the HDL receptor, namely, the Scavenger Receptor Class B type 1 (SR-B1), in the target cells. SR-B1 and HDL represent crucial mediators of cholesterol metabolism. Herein, we review the complex role of HDL and SR-B1 in SARS-CoV-2-induced disease. We also review recent advances in our understanding of HDL structure, properties, and function during SARS-CoV-2 infection and the resulting COVID-19 disease.

## 1. Introduction

The novel severe acute respiratory syndrome coronavirus 2 (SARS-CoV-2) is the etiologic agent of the severe respiratory illness called coronavirus disease 2019 (COVID-19) [1,2], which is a transmissible respiratory disease affecting 223 countries and, as of August 2021, accountable for more than 200 million confirmed cases and around 4.5 million confirmed deaths worldwide [3]. SARS-CoV-2 is a positive-sense single-stranded, non-segmented, enveloped RNA virus that utilizes endocytosis or membrane fusion to enter host cells [4]. The first step in viral entry into the host cell is the binding of the viral transmembrane spike (S) glycoprotein to a host cell transmembrane angiotensin-converting enzyme 2 (ACE2) protein [5]. Subsequently, membrane proteases, such as furin and the transmembrane protease serine 2 (TMPRSS2), sequentially cleave the spike protein, thereby priming it for entry [6]. However, other cellular proteases, such as cathepsins B and L, have also been implicated in SARS-CoV-2 spike priming [7]. Following host cell infection, ACE2 is downregulated and its enzymatic activity on the membrane is reduced [8]. Some studies showed that the mucosa of the oral cavity [9] and lung epithelial cells [10] highly express ACE2, which determines the initial tissue entry site of the virus. Endothelial cells [11], as well as human heart pericytes [12], also have high expression of ACE2, which may explain the pathophysiological effects related to the cardio-pulmonary system [13,14,15,16].

Studies of human clinical samples showed that SARS-CoV-2 viral infection is associated with changes in the host plasma lipid profile, which were proposed as a potential biomarker to support diagnostics and clinical management [17,18,19,20]. Multiple studies showed reduced high-density lipoprotein-cholesterol (HDL-C) levels, reflecting reduced levels of HDL particles, in the plasma of COVID-19 patients compared to healthy subjects [17,18,21,22]. These studies demonstrated an inverse correlation between HDL levels and the severity of COVID-19, with low HDL levels being associated with an increased risk of severe outcomes [17,18,21,22]. Furthermore, low levels of HDL were associated with decreased SARS-CoV-2 clearance [23], while patient recovery was accompanied by a correction in HDL-C and other lipid parameters [18,21,24]. It was reported that reduced levels of HDL in COVID-19 patients were associated with increased levels of the inflammatory marker C-reactive protein (CRP) [18,21,22,24] and possibly development of the “cytokine storm” that was observed in certain COVID-19 patients [19,23,24,25]. The above observations are consistent with HDL exerting a protective effect in COVID-19 disease, although the alternative, namely, that low HDL may be an epiphenomenon, cannot be ruled out. However, it was reported that SARS-CoV-2 binds to HDL particles via the virus’s spike protein and in vitro studies using human embryonic kidney 293 (HEK293) and human hepatoma HepG2 cell lines have shown, at least under certain conditions (such as low HDL particle concentrations), that this facilitates SARS-CoV-2 binding to ACE2 and the infection of cells [26,27]. On the other hand, others have reported findings with cultured human hepatoma Huh-7 and African green monkey kidney Vero-E6 cell lines, demonstrating that, while low concentrations of HDL promote SARS-CoV-2-mediated infection, higher HDL particle concentrations suppress SARS-CoV-2 infection (compared to the absence of HDL) [27,28]. Furthermore, both the enhancement of SARS-CoV-2 infection by low concentrations and the suppression of SARS-CoV-2 infection by high concentrations of HDL particles appears to be dependent on the expression of the HDL receptor, namely, the scavenger receptor class B type I (SR-B1, encoded by the *SCARB1* gene) [29,30], in the target cells [26,28]. These findings suggest that HDL particles may exert complex effects on COVID-19 disease. To try to shed light on the complex role of HDL in SARS-CoV-2-induced disease, we reviewed recent advances in our understanding of HDL structure, properties, and functions during SARS-CoV-2 infection and the resulting COVID-19 disease.

## 2. Virus Infection and Cholesterol

Cholesterol is an essential lipid component of biomembranes of animal cells and enveloped animal viruses (reviewed in [31,32]), including coronaviruses [33,34,35,36,37,38]. Cholesterol-rich microdomains, called lipid rafts, on host cell plasma membranes have an important role in viral entry and budding [39]. Cholesterol-depleting molecules, such as methyl-β-cyclodextrin, inhibit the cellular entry of several viruses, such as human immunodeficiency virus type 1 (HIV-1), rotaviruses, and coronaviruses [38,40,41,42]. Mechanistic studies of HIV infection showed that cholesterol promotes proper spatial organization and interaction of host receptors and co-receptors within lipid rafts facilitating viral entry [37]. A recent study demonstrated that plasma membrane cholesterol increased the trafficking of ACE2 and the furin protease to lipid rafts, thereby increasing SARS-CoV-2 infection [22]. Specifically, the removal of plasma membrane cholesterol (by treatment with methyl-β cyclodextrin) reduced the levels of ACE2 and the furin protease in lipid rafts, thereby reducing SARS-CoV-2 infection; on the other hand, loading cells with cholesterol via treatment with apolipoprotein (Apo) E and serum increased the trafficking of ACE2 and the furin protease to lipid rafts, resulting in increased SARS-CoV-2 infection [22]. This suggests that plasma membrane cholesterol is required for proper trafficking and localization of receptors that facilitate SARS-CoV-2 infection. This study shows the importance of cholesterol and cholesterol-rich microdomains of the host plasma membrane for the proper assembly of SARS-CoV-2 entrance factors.

Some viruses were shown to use cellular receptors that are involved in cholesterol homeostasis to infect target cells. For example, the hepatitis C virus co-opts the low-density lipoprotein receptor (LDLR) and the HDL receptor SR-B1 to infect hepatocytes. As mentioned above, it was recently demonstrated that SR-B1 may affect the ability of SARS-CoV-2 to infect cells by mediating the effects of HDL on virus infection [26,27,28].

## 3. The HDL Receptor SR-B1

SR-B1 (encoded by the *SCARB1* gene) is a multi-ligand scavenger receptor with a high affinity for HDL and mediates the selective exchange of lipids between cells and bound HDL particles [43,44] (Figure 1). SR-B1 is a member of a family of structurally related receptors, including SR-B2 (CD36) and SR-B3 (LIMP2), in mammals. Each of these contains two transmembrane domains, namely, a relatively short cytoplasmic amino and carboxy termini and a relatively large and heavily glycosylated extracellular (lysosomal in the case of LIMP2) domain through which a hydrophobic channel extends to the membrane-proximal region [44]. It is apparently through this channel that SR-B1 mediates the exchange of lipids between the cell membrane and the HDL bound to its extracellular domain (Figure 1).

By virtue of its ability to mediate the lipid exchange between cells and bound HDL, SR-B1 plays a key role in the accumulation of cholesterol in steroidogenic cells, such as those of the adrenal cortex, which store HDL-derived cholesterol in cytoplasmic lipid droplets for use in steroid hormone biosynthesis [44,48] (Figure 2). Likewise, SR-B1 in hepatocytes plays a key role in mediating the cellular uptake of HDL-derived cholesterol, which is either secreted (as cholesterol or after conversion to bile acids) into the bile for elimination or repackaged into nascent lipoproteins and secreted into the blood as part of the cholesterol cycle [44,48]. In this way, SR-B1 drives a process called reverse cholesterol transport (RCT) through which HDL clears excess cholesterol from the artery wall and carries it to the liver [44,48] (Figure 2). RCT is believed to be a major pathway by which HDL protects against the development of atherosclerotic cardiovascular disease [49,50,51]. Studies in mice showed that the absence of functional SR-B1 leads to increased plasma cholesterol concentrations, which is associated with enlarged HDL particles with increased unesterified cholesterol content and altered apolipoprotein composition [52,53,54,55,56]. Although inactivating mutations in the *SCARB1* gene in humans are rare, similar increases in HDL cholesterol as well as early-onset coronary artery atherosclerotic disease were reported in human compound heterozygotes carrying two inactivating mutations in *SCARB1* [57]. This is similar to findings in mice in which SR-B1 is knocked out in the context of other atherogenic mutations [54,58,59]. This highlights the important atheroprotective role that is played by SR-B1-mediated HDL-dependent RCT.

## 4. SR-B1 and Viral Infection

SR-B1 was previously reported to mediate hepatitis C virus (HCV) entry into hepatocytes [60,61] and to enhance dengue virus infection [62]. Wei and colleagues recently demonstrated that, in the presence of low concentrations of HDL, SR-B1 facilitates SARS-CoV-2 infection of cultured human hepatoma Huh-7 cells and African green monkey Vero E6 cells. They reported that in the presence of 6 µg HDL protein/mL, SARS-CoV-2 infection was enhanced by SR-B1 overexpression and inhibited by SR-B1 silencing or pharmacological inhibition [26]. They showed that while SR-B1 cannot bind to the virus spike protein directly, the virus spike protein mediated the SARS-CoV-2 binding to HDL particles, and it appears that SR-B1 binding to those HDL particles enhanced SARS-CoV-2 infection of cells via ACE2. Finally, an antibody against the S1 subunit of the spike protein that reportedly blocks the putative cholesterol/HDL-binding site strongly inhibited the entry of pseudoviral particles into the cell [26]. In the case of HCV infection of hepatocytes, different ligands of SR-B1 were shown to differentially affect infection of cultured cells: HDL was also shown to enhance, while oxidized LDL (another SR-B1 ligand which may bind to a different ligand-binding site from that engaged by HDL) inhibited HCV infection [63,64]. Furthermore, PDZK1, which is an adaptor protein that binds to SR-B1’s carboxy-terminal cytoplasmic tail and stabilizes it against degradation in liver hepatocytes [65], was found to be essential for the efficient entry of HCV into cultured hepatocyte-like cells [66]. This was most likely because the deletion of PDZK1 in liver hepatocytes resulted in a dramatic loss of SR-B1 protein [67]. Furthermore, small-molecule inhibitors of SR-B1-mediated lipid transfer (blockers of lipid transfer BLT-3 and BLT-4) were reported to abrogate the stimulation of HCV infectivity by human serum or HDL, suggesting that the enhancement of viral infection might be dependent on SR-B1’s lipid exchange activity [68,69]. Similarly, Wei and colleagues reported that BLT-1, which is a related small-molecule inhibitor of SR-B1-mediated lipid transfer between bound HDL and cells, inhibited the SARS-CoV-2 infection of Huh-7 hepatocyte-like cells [26]. This suggests that SR-B1-mediated lipid transfer activity may be necessary for the HDL stimulation of SARS-CoV-2 infection.

Cho and co-workers also reported data showing that low concentrations of HDL (15 µg protein/mL) enhanced the SARS-CoV-2 infection of cultured Vero E6 cells, consistent with the findings of Wei et al. [26,27]. These low concentrations (6–15 µg HDL protein/mL) are much lower than those normally found in plasma (approximately 1000–1500 µg HDL protein/mL [70]) or expected for interstitial fluids (typically 10–25% of plasma concentrations [71]). However, Cho and co-workers also reported that higher HDL concentrations (in the order of 60 µg protein/mL) suppressed viral infection [27], although it was suggested that differences in the HDL preparations may have contributed to the differences between their findings and those reported by Wei and co-workers [72]. Similarly, Thaxton and co-workers reported (in a preprint) that a synthetic HDL-like nanoparticle also suppressed SARS-CoV-2 infection of cultured HEK293 and Vero E6 cells in a manner that was suppressed by antisense-mediated SR-B1 knockdown [28]. Together, these results may suggest that while low (sub-physiological) concentrations of HDL appear to enhance SARS-CoV-2 infection of cultured cells, higher HDL concentrations appear to suppress SARS-CoV-2 infection of cultured cells. In both cases, this involved pathways that are dependent on the HDL receptor SR-B1. Whether the suppression of SARS-CoV-2 infection of cells, mediated by higher concentrations of HDL, is dependent on SR-B1’s lipid transfer activity and whether these processes occur in vivo in animal models or human patients remains to be determined.

As mentioned above, both ACE2 and SR-B1 were reported to localize (at least in some cell types and under certain conditions) to cholesterol-rich domains (lipid rafts/caveolae) [22,73]. Interestingly, SR-B1 is known to modify the cholesterol content of the plasma membrane. For example, when overexpressed in cultured Chinese hamster ovary (CHO) and COS-7 fibroblast-like cells, SR-B1 increases the accessibility of cholesterol on the outer leaflet of the plasma membrane, as measured by the cholesterol’s sensitivity to modification by cholesterol oxidase [74,75,76]. This may be the consequence of the SR-B1-mediated transfer of cholesterol from HDL particles into the plasma membrane, increasing local cholesterol concentrations, or SR-B1-mediated transfer of cholesterol from the membrane to HDL particles, reducing the local concentration, since SR-B1 can mediate the exchange of cholesterol between cells and bound HDL particles in both directions [76,77] (Figure 1). In either case, this is dependent on SR-B1’s hydrophobic lipid channel, which is blocked by the BLT class of inhibitors [47,78]. On the other hand, it was reported that SR-B1-mediated redistribution of plasma membrane cholesterol may be independent of its ability to bind HDL [74,75].

Nevertheless, this suggests a possible mechanism by which SARS-CoV-2 binding to HDL may enhance its infection of cells: due to the virus binding to HDL, the HDL binding to SR-B1, and the virus binding to ACE2, this may bring SR-B1 into proximity to ACE2 on the cellular plasma membrane, resulting in an SR-B1-mediated redistribution of cholesterol in the local environment of ACE2 in such a way as to favor viral entry. Circumstantial evidence supports this model. For example, SR-B1, ACE2, and furin were all reported to be present in cholesterol-rich microdomains on the plasma membrane [22,73]. Cholesterol-rich microdomains were suggested to function as entrance gates for viruses [34,41,79], including SARS-CoV-2 [80]. The depletion of membrane cholesterol from lipid rafts with cyclodextrin interfered with the localization of ACE2 and furin in lipid rafts and reduced SARS-CoV-2 infection [22]. The ability of SR-B1 to modulate plasma membrane cholesterol distribution/content suggests that it may influence the cholesterol content of lipid rafts and the localization of lipid-raft-associated proteins, such as ACE2 and furin. Therefore, it is possible that through spike protein binding to cholesterol, SARS-CoV-2 association with HDL may position it to exploit the physiological function of SR-B1 to optimize conditions for its fusion and entry processes within host cells.

However, this explanation does not easily explain the apparent HDL concentration dependence of HDL’s effect on SARS-CoV-2 infection of cells [28]. The ability of SARS-CoV-2 to bind to HDL (via the spike protein binding to cholesterol), as well as the relative sizes of spherical HDL particles (up to 14 nm in diameter [81]) and SARS-CoV-2 particles (100 nm in diameter [82]) suggests that low HDL concentrations may favor a single virus particle binding to a single HDL particle via the virus particle spike proteins binding to cholesterol on the HDL particle, but leaving abundant virus spike proteins accessible for interaction with the cell surface ACE2 (see Figure 3). This would also allow for bridging between SR-B1 (via HDL binding) and ACE2 (via SARS-CoV-2 binding), which is consistent with enhanced SARS-CoV-2 infection mediated by low HDL concentrations [26]. On the other hand, higher, more physiological HDL concentrations may favor multiple HDL particles binding to individual SARS-CoV-2 particles (Figure 1). Since this binding occurs via SARS-CoV-2 spike proteins binding to HDL cholesterol [26,27,28], the decoration of virus particles with HDL particles may prevent the virus spike proteins from interacting with ACE2 receptors on the cell’s surface. Alternatively, the concentration of many HDL particles in the vicinity of ACE2 receptor may alter plasma membrane cholesterol in such a way as to make viral entry, or other steps in virus infection, less favorable.

Cell lysis is often used as a surrogate for virus infection in cell culture studies [27]. Therefore, possible effects of HDL on other steps in the virus life cycle, such as viral replication or budding from cells, should be considered. For example, in addition to viral entry, lipid rafts also play a role in the budding of some viruses from cells. HIV and rotavirus proteins were shown to sort into lipid rafts [83,84,85,86]. Furthermore, cholesterol found on viral envelopes is derived from the cell membrane from which the virus buds and plays an important role in viral stability and infectivity. The depletion of cholesterol from viral envelopes of some viruses reduces infectivity and cholesterol replenishment restores infectivity [33,79,87]. Analogous to its role on host cell membranes, cholesterol on viral envelope membranes was shown to play a role in organizing the receptors required for viral fusion, as seen in the assembly of hemagglutinin trimers on the Influenza viral envelope [88]. Therefore, HDL and/or SR-B1 may also impact viral infection indirectly by regulating plasma membrane cholesterol distribution and the cholesterol content of viral envelopes. Whether this is relevant for SARS-CoV-2 is not yet known. Given the complex effects of HDL on SARS-CoV-2 infection of cells revealed from in vitro cell culture studies, it is important to understand the nature of HDL particles in both healthy individuals and those infected with SARS-CoV-2.

## 5. HDL Structural Complexity and Function

HDL is characterized by a small particle size (7–14 nm diameter), high density (1.063–1.21 g/mL), and unique apolipoprotein composition [89,90], distinguishing it from other lipoproteins. Proteomic analysis of HDL revealed over 200 associated proteins involved in multiple processes, including lipid transport, inflammation, immunity, hemostasis, and proteolysis [91,92,93,94,95]. This large number of proteins, together with the small particle size, emphasizes that HDL represents a class of diverse lipoprotein particles with diverse functions despite similar physical characteristics [91,92,93,94]. Proteomic analysis led to the identification of distinctive proteome profiles that are associated with distinct HDL sub-species, suggesting that distinct HDL functions may be attributed to particular HDL sub-species. However, the relationship between HDL function and HDL sub-species is not fully understood [89]. HDL lipids consist mainly of phospholipids, sphingomyelin, and unesterified cholesterol at the particle surface with cholesteryl esters and small amounts of triglycerides comprising a hydrophobic core [96,97]. In addition, lipidomic analyses revealed the presence of a range of minor lipid species that may contribute to the functional diversity of HDL particles. These include lysophospholipids, such as sphingosine-1-phosphate (S1P); ceramide; plasmalogens; bioactive steroids, such as oxysterols and estrogen; and minor amounts of di- and mono-acylglycerol and free fatty acids [98]. HDL particles were also reported to carry lipid-soluble vitamins, such as vitamin E [99], and other bioactive macromolecules, including microRNAs [100]. Furthermore, the HDL particle composition is dynamic, changing with different physiological and pathophysiological states [94,98,101]. Together, this diversity in composition gives rise to substantial variability and complexity in the HDL particle’s function.

Despite the variety of HDL particles, their biosynthesis begins with the synthesis and secretion of the major apolipoprotein (ApoA1) as a lipid-free protein, which acquires phospholipids and cholesterol in the circulation through efflux mediated by ATP-binding cassette transporter (ABC)-A1 to form disc-shaped premature HDL particles [51,102,103,104,105] (Figure 2). These particles continue to accumulate unesterified cholesterol, which, through the action of lecithin:cholesterol acyltransferase (LCAT), is converted into the more hydrophobic cholesteryl ester that is sequestered in the center of HDL, forming a hydrophobic core, ultimately shaping HDL into a spherical particle [106,107]. Over time, HDL particles increase in size as they circulate through the bloodstream and incorporate more cholesterol from cells via ABCG1-mediated cholesterol efflux, and exchange lipids with other lipoproteins through the actions of phospholipid transport protein (PLTP) and cholesteryl ester transfer protein (CETP) (reviewed by [49]) (Figure 2).

HDL cholesterol (both free and esterified) is taken up by hepatocytes in the liver via a process called selective lipid uptake in which the lipid components of the HDL particle are taken up by the cell without the net internalization and lysosomal degradation of the lipoprotein particle itself [44,108,109], distinguishing the mechanism of HDL cholesterol delivery to cells from the better known endocytic mechanisms involved in the delivery of cholesterol by other lipoproteins, such as low-density lipoprotein. A similar pathway of selective lipid uptake from HDL occurs in steroidogenic cells of the adrenal gland, testes, and ovary [44,108,109]. The identification of SR-B1 as an HDL receptor led to the discovery that it mediates selective HDL lipid uptake and is responsible for the pathway of cholesterol delivery to the liver and steroidogenic tissues [43,44]. HDL-mediated selective cholesterol uptake by hepatocytes in the liver is the final step in the athero- and cardio-protective RCT pathway in which HDL picks up excess cholesterol from the artery wall and delivers it, as described above, to the liver for recycling or biliary excretion [44,48] (Figure 2).

## 6. Anti-Oxidative and Anti-Inflammatory Properties of HDL

HDL is recognized to have pleiotropic cellular effects, such as anti-oxidant [110,111], anti-inflammatory [110,112,113], and anti-cytotoxic effects [114,115,116,117,118,119,120]. Moreover, it is involved in the cholesterol efflux of cells [50,97], which protects macrophages from LDL-induced apoptosis and enhances endothelial function by activating nitric oxide synthase to promote endothelial repair and induce angiogenesis [121,122,123]. Cholesterol efflux capacity mainly depends on HDL’s size, HDL acting as an extracellular acceptor, macrophage cholesterol content, and the expression of macrophage cholesterol transporter proteins [124,125]. In vivo studies indicated that ABCA1 and ABCG1 play a key role in facilitating cholesterol efflux and reverse cholesterol transport [126,127]. Moreover, mice that are deficient in these transporters have marked increases in foam cell accumulation and display enlarged cholesterol-rich lipid rafts with high toll-like receptor 4 expression and an elevated response to stimulation by LPS [128,129,130].

HDL also reduces the expression of cytokines, such as tumor necrosis factor α (TNF-α) and interleukin-1 that mediate the upregulation of leukocyte–endothelial adhesion molecules. This process is believed to be mediated by ApoA1, as well as HDL-associated lipids, including S1P [131,132]. The majority of S1P in plasma was reported to be carried by HDL, bound to HDL-associated ApoM [133]. ApoM was reported to be present on only 5% of HDL particles [134], suggesting that only a subset of HDL particles carry S1P and may represent a sub-species that is responsible for at least some of HDL’s protective functions. It was shown that the S1P receptor 1 (S1P1) is responsible for the anti-inflammatory effects of HDL-associated S1P and induces a shift in macrophage polarization from the pro-inflammatory M1 to the anti-inflammatory M2 phenotype [135]. HDL-associated S1P and S1P1 also stimulate phosphatidylinositol-3-kinase (PI3K)/Akt and signal transducer and activator of transcription-3 (STAT3) survival signaling pathways, leading to macrophage chemotaxis and protection against apoptosis [114,120,135,136]. Furthermore, HDL-associated S1P acts as a biased ligand for S1P1, suppressing inflammatory activation of endothelial cells [137].

The genetic manipulation of ApoA1 expression in septic mice showed that its deficiency strongly increased inflammatory cytokine production, while ApoA1 overexpression in septic mice, or their treatment with HDL or with ApoA1 mimetic peptides, reduced inflammation and protected them from sepsis, at least in part, through neutralization of bacterial endotoxins [131,132,138,139,140]. These properties appear to be dependent on the abundance of HDL particles containing ApoA1. Similarly, in a mouse model that was engineered to express CETP, the pharmacological inhibition of CETP, leading to increased HDL-cholesterol levels, suppressed sepsis-induced inflammation and improved survival [141]. In agreement with this, CETP gain-of-function mutations in humans were associated with increased mortality due to sepsis, while reduced CETP function was associated with increased survival [141]. CETP exchanges cholesteryl esters on HDL for triglycerides from other lipoproteins (Figure 2); therefore, CETP inhibition or reduced CETP activity results in increased HDL cholesterol levels, whereas increased CETP function is associated with lower HDL cholesterol. This demonstrates, at least in the setting of sepsis, that HDL-targeted therapy may have beneficial outcomes.

The protein and lipid compositions of HDL particles also influence their anti-oxidative properties [142]. HDL’s ability to decrease LDL oxidation is inversely correlated to its free cholesterol and sphingomyelin content and positively correlated to its S1P content [143]. Furthermore, several HDL apolipoprotein components, such as ApoA1 [143,144], ApoE [144], and ApoM [145], were correlated with HDL anti-oxidative properties [89]. ApoA1 is essential both for the structure of HDL and the maintenance of the lipid environment in which enzymes such as paraoxonase 1 (PON1) and LCAT operate [96]. Therefore, it is likely that ApoA1 plays an important role (albeit indirect) in the anti-oxidative properties of HDL.

HDL-associated hydrolases, including PON1, LCAT, and platelet-activating factor acetylhydrolase (PAF-AH), contribute strongly to the anti-oxidative activity of HDL [146,147]. There are three members of the paraoxonase family: PON1, PON2, and PON3. PON1 and PON3 are synthesized by the liver and secreted into the blood, where they assemble with HDL particles [148,149]. PON1 can hydrolyze a wide variety of substrates, such as lactones, glucuronide drugs, thiolactones, aryl esters, cyclic carbonates, organophosphorus pesticides, and nerve gases [150]. Knocking out PON1 in mice results in HDL with impaired anti-oxidative activity [151,152]. An interesting recent work by Huang et al. [153] proposed that PON1 can form a tertiary complex with HDL and myeloperoxidase, which can modulate the activity of the latter. LCAT can hydrolyze oxidized acyl chains of oxidized phospholipids, generating lysophosphatidylcholine and oxidized free fatty acids [154]. Furthermore, LCAT can work as a chain-breaking anti-oxidant via its cysteine residues [155]. There is evidence that mutations in LCAT may reduce anti-oxidative capacity of HDL [156]. However, the inhibition of LCAT activity does not affect HDL’s capacity to neutralize lipid hydroperoxides derived from LDL [157].

## 7. Effects of Chronic Disease on HDL Structure and Function and Implications for COVID-19

Abundant research over the past decade has revealed that HDL’s composition and function change dramatically with various chronic diseases, including infectious diseases, such as sepsis; metabolic diseases, such as obesity and diabetes; and cardiovascular disease [91,158,159,160,161,162,163]. This includes changes in the levels of HDL-associated proteins and lipid components and/or modification of those components, such as by non-enzymatic glycation and oxidation of both protein and lipid species, leading to the accumulation of dysfunctional HDL particles [158] (Figure 4).

In general, cardiovascular and inflammatory diseases are accompanied by decreases in some HDL-associated apolipoproteins, including ApoA4, and increases in other proteins, such as the acute phase protein serum amyloid A-1 (SAA1) [91,159]. Inflammatory HDL is characterized by a replacement of the major protein constituent of HDL, namely, ApoA1, with the acute phase protein SAA1 [159,160,161,164]. SAA1-enriched HDL particles lead to higher HDL binding to macrophages, a reduction in cholesterol efflux by macrophages, and increased selective uptake of CE by macrophages [161,165]. This suggests that the alterations in HDL properties as a result of changes that are brought about by chronic inflammation may shift HDL from protecting to promoting disease development (Figure 4).

Chronic metabolic disease was also reported to lead to the modification of HDL-associated ApoA1. For example, HDL from type 2 diabetes (T2D) patients contains ApoA1 and ApoA2 proteins that are modified by non-enzymatic glycation [162,163,166,167]. Glycation leads to multimerization of ApoA1 and ApoA2, forming high-molecular-weight aggregates [168,169]. The glycation of ApoA1 may impair its ability to stimulate cholesterol efflux from cells, as well as its ability to activate LCAT, which plays an important role in HDL maturation and its anti-oxidative properties [170]. Glycated ApoA1 has a three-fold shorter half-life than native ApoA1, leading to accelerated turnover/degradation [171,172]. Finally, the glycation of HDL was reported to compromise its anti-inflammatory properties [173]. Similar modifications were reported in HDL from elderly subjects and HDL from smokers [169,171,174,175,176,177]. Oxidation of HDL-associated ApoA1 was also reported in those with metabolic disease, the elderly, and in smokers, and dramatically reduces HDL’s ability to mediate cholesterol efflux [178,179,180]. HDL from diabetic patients and smokers also exhibits elevated triglyceride levels compared to non-diabetics and non-smokers, respectively [181,182,183,184]. Similarly, SAA-enriched HDL from myocardial infarction patients contained increased triglyceride content [160] (Figure 4). At least one report suggests that chronic disease may impact HDL properties in such a way as to influence its effects on SARS-CoV-2 infection of cells. Specifically, Cho and co-workers reported that while native HDL (60 µg protein/mL) could suppress SARS-CoV-2 infection of cultured Vero E6 cells, the glycation of HDL (as seen in patients with type 2 diabetes) impaired its ability to suppress SARS-CoV-2 infection [27]. However, the effect of HDL glycation on the ability of low concentrations of HDL to enhance SARS-CoV-2 infection of target cells has not been described. Furthermore, several studies reported interactions between obesity, diabetes, smoking, and other cardiometabolic risk factors with disease severity in COVID-19 patients [185,186,187,188,189]. However, these relationships were reported to be no longer significant when HDL was controlled for, suggesting that the effects of these conditions on COVID-19 may be mediated through HDL. In fact, one study (unpublished at the time of this writing) suggests that baseline HDL levels appear to be associated with the risk of hospitalization for COVID-19 [190]. Therefore, the quality and function of HDL in individuals may influence the outcomes of subsequent SARS-CoV-2 infection.

## 8. HDL Alterations in COVID-19

Several reports showed that, consistent with other chronic and acute inflammatory conditions, dramatic changes in the abundance and composition of HDL particles occur during COVID-19 disease development [191] (Figure 4). These changes include substantial reductions in overall HDL-cholesterol levels, and in HDL-associated apolipoproteins including ApoA1; ApoA2; ApoA4; ApoC-1, 2, 3; ApoJ; ApoE; ApoD; ApoF; PON1; and PON3 [191,192]. A recent study showed that the serum levels of S1P and its HDL-associated carrier ApoM are also significantly decreased in COVID-19 patients compared to healthy volunteers [191,193]. On the other hand, SAA1 is increased on the HDL of COVID-19 patients in a manner that was associated with COVID-19 severity [191,194]. The increased association of SAA1 with HDL was also accompanied by a blunting of the anti-apoptotic and anti-inflammatory effects of HDL examined in cultured human umbilical vein endothelial cells (HUVEC) that were challenged with TNF-α [191]. It was suggested that HDL-bound SAA may be useful as a biomarker for COVID-19 severity and prognosis [194]. Although HDL was reported to carry a variety of microRNA species and to act to shuttle microRNAs between different cell types and tissues, the effects of COVID-19 on HDL-associated microRNAs have not yet been reported. Whether/how the observed changes in HDL structure, composition, and function during COVID-19 disease affect the ability of HDL to modify SARS-CoV-2 infection of target cells has not been explored. As mentioned above, it was reported that native HDL is able to affect SARS-CoV-2 infection of cultured Vero E6 cells in a dose-dependent manner [27]. It would be of interest to determine whether HDL enriched in SAA1 and depleted of ApoA1, as seen in patients with COVID-19 and other inflammatory diseases, may exhibit an altered ability to modify SARS-CoV-2 infection of target cells. If so, this may suggest a mechanism whereby ongoing disease promotes virus infection by impairing the protective effects of HDL. Furthermore, the observed reductions in HDL (particularly ApoA1-containing HDL) during COVID-19 disease may favor conditions in which low HDL levels actually promote virus infection of target cells, as was reported in cell culture systems [26,27]. Further experiments are required to determine whether this may be the case. For example, if the reductions in HDL levels that occur during the course of COVID-19 disease impact on the severity of disease, then interventions that are aimed at raising HDL levels may be beneficial. In this light, the recent reports that CETP inhibition is beneficial in sepsis, at least in mouse models, suggests similar interventions may have beneficial impacts on COVID-19 disease [141].

## 9. Spike Protein, HDL, and Endothelial Cell Function

Emerging evidence in the literature suggests that SARS-CoV-2 and its spike protein promote endothelial cell toxicity [195,196,197]. Indeed, it is possible for SARS-CoV-2 to enter the circulation and affect multiple tissues and cell types expressing ACE2 [10], including endothelial cells [198]. Endothelial dysfunction is increasingly becoming a concern as it is believed to be at the core of several life-threatening complications associated with COVID-19, such as venous thromboembolic disease and multi-organ involvement [199].

Histological analysis of tissue samples from deceased COVID-19 patients showed systemic vascular disease associated with thrombi in microvessels and endothelial damage/necrosis [196]. Histological analysis of lung tissue showed an accumulation of the virus and evidence of apoptosis in capillary endothelial cells [196]. Moreover, histological examination of lung capillaries showed marked microangiopathy, as displayed by the fragmented appearance of the capillary walls and disruption in capillary endothelial cell adhesion. The extrapulmonary tissues examined displayed a significantly lower SARS-CoV-2 viral RNA signal compared to that observed in the lung tissue but did show strong staining of SARS-CoV-2 pseudovirions (SARS-CoV-2 spike/capsid proteins lacking RNA), which co-localized with ACE2-positive endothelial cells of microvascular beds. These pseudovirion-positive endothelial cells displayed an upregulation of complement proteins, pro-inflammatory cytokines, and caspase-3 staining. In response to these findings, Magro and colleagues proposed a model for explaining SARS-CoV-2 disease sequelae, where (1) initial microangiopathy in infected lung capillaries causes endothelial cell death releasing pseudovirions into the circulation, and (2) the pseudovirions dock on ACE2-positive endothelial cells of extrapulmonary microvascular beds, leading to endothelial toxicity, activation of the complement pathway resulting in a pro-coagulant state, and cytokine release culminating in a cytokine storm. These findings were further supported in a follow-up study that examined the colocalization of spike proteins, cytokines, complement proteins, and caspase-3 in HUVEC cells in vitro following incubation with spike proteins, and in endothelial cells of microvascular beds in vivo following the injection of spike protein in mice [200]. Other groups showed that the SARS-CoV-2 spike protein alone can cause endothelial damage/dysfunction manifested via impairments in endothelial nitric oxide synthase activity and mitochondrial function [201]. Finally, another study conducted a histological analysis of COVID-19 tissue specimens and showed viral inclusions in endothelial cells, endothelial inflammation, and endothelial apoptosis, as indicated by caspase-3 staining [195]. Altogether, there is compelling evidence of the vasculopathy associated with SARS-CoV-2 infection.

The SARS-CoV-2 spike protein was also implicated as a mediator of “long COVID,” which is a term that is used to describe the long-term persistence of symptoms in COVID-19 patients following recovery from acute virus infection and discharge from the hospital [202]. The symptoms of long COVID vary but include shortness of breath, cough, fatigue, and depression, and reflect the persistent effects on multiple organs [202]. Much remains unknown regarding the causes of long COVID; however, there was some suggestion that the SARS-CoV-2 spike protein alone can trigger persistent alterations in cellular inflammatory gene expression in cultured human primary bronchial epithelial cells [203], suggesting a potential mechanism that may be involved. However, whether and how HDL and SR-B1 might affect this is not known.

It is well known that HDL enhances endothelial function, promotes endothelial cell integrity, and suppresses inflammatory gene activation in endothelial and other cell types [137,204,205,206]. This stems from HDL’s anti-inflammatory, anti-oxidative, and anti-apoptotic effects, possibly reflecting the roles of specific sub-species of HDL particles, resulting in the activation of signaling pathways stemming from their interaction with SR-B1 and HDL-bound S1P-activating S1PRs [137,204,205,206]. The SARS-CoV-2 spike protein is known to interact with HDL, at least in vitro, by virtue of its ability to bind cholesterol [26,207]. If it similarly interacts with HDL in vivo, then this may further contribute to the alterations in HDL structure and function, impacting HDL’s ability to elicit protective signaling in endothelial and other cell types, preventing normal HDL-mediated protective pathways, and rendering cells susceptible to processes triggering inflammation and dysfunction. Furthermore, the association of SARS-CoV-2 spike protein with HDL may target the spike protein to cell types that are normally protected by HDL, such as endothelial cells, which express the HDL receptor SR-B1. Thus, HDL may serve to facilitate spike-protein-induced cellular toxicity. On the other hand, it is possible that the binding of a spike protein by HDL may serve to sequester and/or clear the spike protein in a fashion that is similar to the HDL-mediated clearance of bacterial endotoxin [140,208,209,210]. Further research is required to understand the consequences of spike protein interactions with HDL and the role of HDL in spike-protein-induced cellular toxicity.

## 10. Conclusions

HDL has the potential to influence disease development resulting from SARS-CoV-2 infection by virtue of its ability to directly modulate infection of cells by the virus, as well as by virtue of its ability to modulate protective functions, such as anti-inflammatory, anti-cytotoxic, and anti-oxidative effects. These protective functions are mediated by HDL sub-species with particular compositions of proteins, lipids, and other macromolecules, of which, ApoA1, ApoM, and S1P are notable examples. The mechanism by which HDL can, under different conditions, either enhance or suppress SARS-CoV-2 infection of cells is not currently known. We have suggested that HDL concentration may be a factor; however, the role of HDL sub-species has not been examined in detail. However, it is important to point out that while associations of HDL levels with COVID-19 disease severity, as well as in vitro cell biological data, may be consistent with a potential for HDL to influence SARS-CoV-2-mediated disease, they do not conclusively show in vivo causality. This awaits in vivo pre-clinical and, if warranted, clinical intervention studies. While HDL may have the potential to influence SARS-CoV-2-mediated disease, the evidence clearly demonstrates that SARS-CoV-2-mediated disease also dramatically influences HDL abundance, composition, and undoubtedly function, similar to the effects of other inflammatory and cardiometabolic diseases. How these two processes intersect during SARS-CoV-2-mediated disease development remains to be further elucidated. Understanding the particular properties and sub-species of HDL that protect against versus promote SARS-CoV-2 infection and resulting disease development may provide opportunities for beneficial therapeutic interventions.

## Figures and Tables

**Figure 1 ijms-22-10182-f001:**
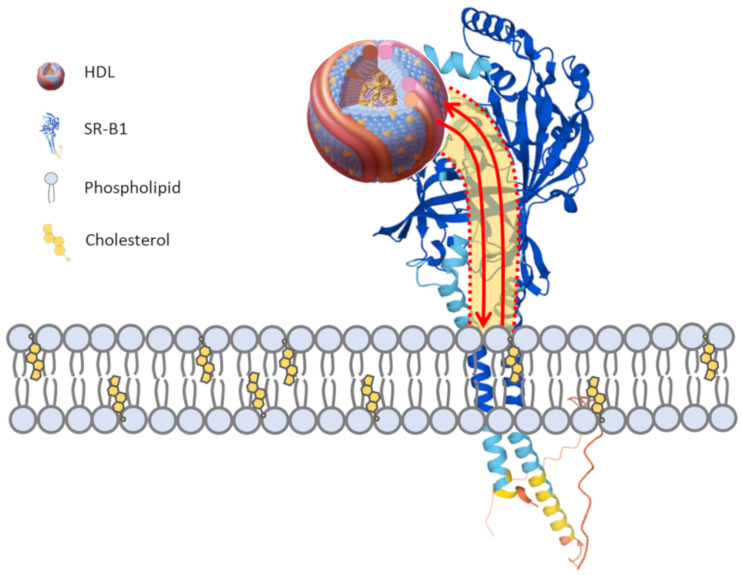
**HDL and SR-B1 selective lipid exchange**. The high-density lipoprotein (HDL) receptor, scavenger receptor class B type I (SR-B1), contains two transmembrane domains and a large extracellular domain that contains a hydrophobic channel (depicted in yellow shading bound by dashed red lines) leading from the HDL binding site to the juxtamembrane region. The binding of HDL to SR-B1 on the surface of cells promotes the bidirectional flow of lipids (red arrows) through the hydrophobic channel of SR-B1 to the membrane. Lipids are exchanged between HDL and the cell without the net internalization or degradation of the HDL particle. The concentration gradient of cholesterol between HDL and the cell plasma membrane is thought to dictate whether there is a net cellular uptake or efflux of lipids. The predicted structure of HDL was from [45]. The predicted structure of human SR-B1 was generated using Alphafold and based on [46]. The SR-B1 structure was first reported by [47].

**Figure 2 ijms-22-10182-f002:**
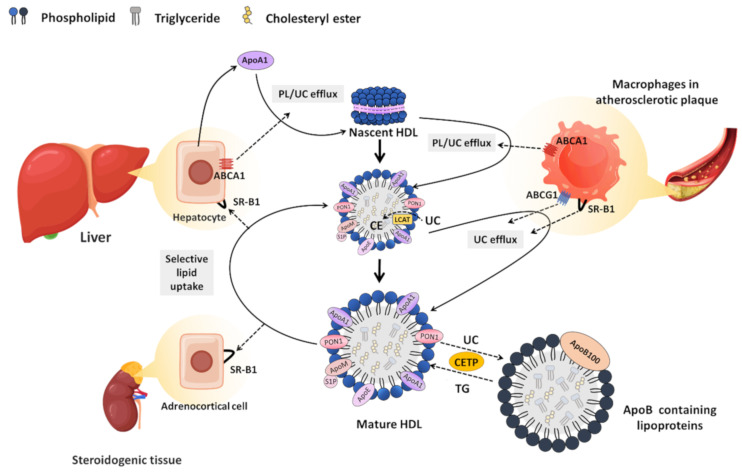
**Biosynthesis and metabolism of HDL.** The biosynthesis of HDL starts with the synthesis and secretion of apolipoprotein (Apo) A-1 as a lipid-free protein from hepatocytes. Through the efflux mediated by the transporter ATP-binding cassette transporter (ABC)-A1, ApoA1 obtains phospholipids (PL) and unesterified cholesterol (UC) from cells to form a disc-shaped nascent HDL particle. These particles take up additional PL and UC through ABCA1-mediated efflux. Lecithin:cholesterol acyltransferase (LCAT) converts UC into the more hydrophobic cholesteryl ester (CE), which is taken to the center of the HDL, shaping HDL into a spherical particle. Spherical HDL serves as an acceptor for UC efflux mediated by ABCG1 and SR-B1. Cholesteryl ester transfer protein (CETP) mediates the exchange of cholesteryl esters in HDL with triglycerides in ApoB-containing lipoproteins, reducing the HDL cholesterol levels. HDL cholesterol (UC and CE) is directly cleared in the liver via SR-B1, which mediates a process called selective lipid uptake in which the lipid moiety of HDL is taken up by cells and the lipid-depleted particle returns to the circulation. Cholesterol taken up in the liver can either be recycled into newly assembled ApoB-containing lipoproteins or can undergo net excretion into the bile. A similar SR-B1-mediated pathway of selective lipid uptake from HDL occurs in steroidogenic cells of the adrenal gland, testes, and ovary to support steroid hormone biosynthesis.Black dotted arrows represent the movement or conversion of cholesterol. Solid black arrows represent the movement or conversion of HDL particles.

**Figure 3 ijms-22-10182-f003:**
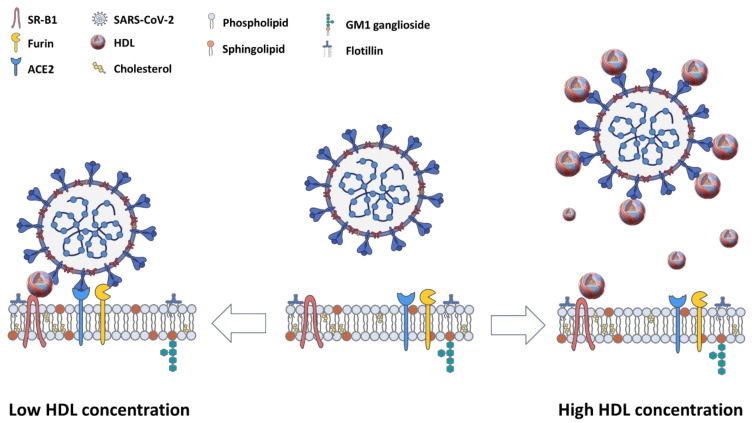
Hypothetical HDL concentration-dependent effects on SARS-CoV-2 infection. In the absence of HDL (**center**), SARS-CoV-2 binds to ACE2 via the virus’s spike protein, but without the participation of the HDL receptor, SR-B1. When HDL is present at a low particle concentration (**left**), HDL particles bind to SARS-CoV-2 via the virus’s spike protein but leave other molecules of spike protein accessible for binding to ACE2 on cells, allowing for bridging of ACE2 and SR-B1 [26]. This may allow SR-B1 to modify the local membrane environment of the ACE2 receptor in a way that favors viral entry, although other mechanisms are possible. When HDL is present at high particle concentrations (**right**), virus particles may bind to multiple HDL particles. This may result in HDL competing with ACE2 for binding to viral spike proteins, preventing SARS-CoV-2 from entry into host cells. Alternatively, the binding of the complex of one virus particle with multiple HDL particles to SR-B1 may result in alterations in the membrane lipid composition that disrupt viral entry into cells. Another possibility is that at higher HDL particle concentrations, HDL that is not bound to SARS-CoV-2 particles may compete with the HDL bound to SARS-CoV-2 for binding to SR-B1, preventing bridging of SR-B1 and the ACE2 receptor. SARS-CoV-2 virus picture was captured from BioRender.com.

**Figure 4 ijms-22-10182-f004:**
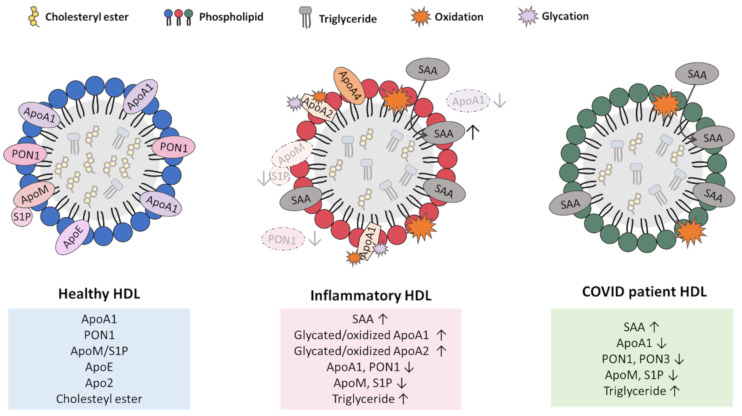
HDL remodeling in inflammatory/cardiometabolic disease and COVID-19. HDL is extensively remodeled during inflammatory and cardiometabolic disease (**center**) and in COVID-19 patients (**right**). Healthy HDL is shown on the **left**. Similar changes in HDL in inflammatory and cardiometabolic disease include oxidation and glycation of HDL-associated proteins, including ApoA1; replacement of ApoA1 with serum amyloid A-1 (SAA1); reductions in paraoxonase 1 (PON1), ApoM/sphingosine-1-phosphate (S1P), and ApoE; and increases in core triglycerides. These changes are associated with reductions in the beneficial functions of HDL (anti-oxidant, anti-inflammatory, and anti-cytotoxic properties, as well as cholesterol efflux) and gain of detrimental functions (pro-oxidant, pro-inflammatory, and pro-cytotoxic properties). Similar changes were reported in HDL during COVID-19 disease, suggesting similar changes in HDL particle functions may contribute to disease development. Arrows pointing up represents increase and arrows pointing down represents decrease for respective compound.

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
