# Peer review of "Good Cholesterol Gone Bad? HDL and COVID-19"

_ijms, 2021, doi:10.3390/ijms221910182_

Round 1

Reviewer 1 Report

It is a well written and well structured review on an interesting and highly relevant topic.

A few comments remain which would improve this manuscript further:

  1. Please note throughout the manuscript better whether described results were obtained in animal models (and which one) or humans. Additionally statements like "cultured cells" need to be clarified: which cells exactly.
  2. Chapter 5 and 6 are rather general. It would be better to only include virus focussed sections as that is the novelty of the current review. Perhaps part of this can be fused with chapter 7/8.
  3. One or multiple introductory figures focussing on chapters 1-3 would be helpfull for a general audience (without detailed background in HDL and lipid metabolism).

Author Response

Point-by-point answers to reviewers

We thank the reviewers for their positive comments about our manuscript and for their helpful and constructive critiques. We enclose, below, a point-by-point response to each of the comments raised by the reviewers:

Reviewer 1    

  1. Please note throughout the manuscript better whether described results were obtained in animal models (and which one) or humans. Additionally, statements like "cultured cells" need to be clarified: which cells exactly.

We thank the reviewer for pointing out this shortcoming. We have now noted throughout the manuscript which results were obtained in animal models and which where obtained in humans and have made note of the cell types utilized for cell culture experimental results.

  1. Chapter 5 and 6 are rather general. It would be better to only include virus focussed sections as that is the novelty of the current review. Perhaps part of this can be fused with chapter 7/8.

While the authors agree with the reviewer that sections 5 and 6 are rather general, we nevertheless feel that a general description of HDL structure, composition and function, is warranted, particularly for those readers less familiar with these topics. We feel that this sets the stage for and will allow the reader to better appreciate the alterations in these properties of HDL that arise with chronic disease, such as COVID-19, which are described in the subsequent sections, 7 and 8.  We note that the reviewer expresses a similar sentiment in their next comment (#3) below. For this reason, we prefer to keep sections 5 and 6 as they are, despite their not being directly related to the virus or the disease caused by it.

  1. One or multiple introductory figures focussing on chapters 1-3 would be helpful for a general audience (without detailed background in HDL and lipid metabolism).

The authors agree with the reviewer. We have therefore introduced two new figures of a general nature for the benefit of a general audience without detailed background in HDL and lipid metabolism. One of these figures (Figure 2) outlines the main processes which are involved in the formation of HDL and by which it serves to mediate reverse cholesterol transport. The other figure (Figure 1) is a diagram of the modeled structure of the HDL receptor illustrating its lipid channel and demonstrating its involvement in the bidirectional transport of cholesterol between bound HDL and the cell membrane. We believe that these two figures be helpful for a general audience without detailed background in HDL and lipid metabolism. As a result of the introduction of these two new figures, the original figures have been re-numbered to Figure 3 and Figure 4.

Reviewer 2 Report

This is an in depth review on HDL and COVID-19 that is generally well written. The flow of concepts is logical and the review can be easily digested.   Minor comments: 1. What is now the official correct nomenclature of Scavenger receptor class B type I (SR-BI) at protein level? It used to be SR-BI and not SR-B1. Did it change? To my knowledge, Monty Krieger insisted that it was SR-BI.  2. Systemic inflammation, as measured by C-reactive protein, is a predictor of venous thromboembolism, acute kidney injury, critical illness, and mortality in COVID-19Therefore, it is not surprising that low HDL cholesterol is  a univariable predictor of venous thromboembolism, acute kidney injury, critical illness, and mortality in COVID-19 or in other words of the severity of COVID-19. Furthermore, many subjects with severe COVID-19 have multiple co-morbidities that result in low HDL cholesterol. Low HDL cholesterol may  an epiphenomenon (when there is no relation with outcomes) or there may be confounding factors (when there is a relationship with outcomes). The in vitro data and cell biological data are consistent with a potential causal role of HDL but this does not prove in vivo causality. 3. If one considers HDL particle concentration as a determinant, one should also consider that the biological response will be dependent on the HDL proteome and lipidome (and mIRNA’s in HDL). 4. All non-standard abbreviations should be explained at first use (e.g. ABCA1, ABCG1,…). These are known by experts in the HDL field but maybe not by other investigators and clinicians dealing with COVID-19. 5. COVID-19 is clearly distinct from sepsis but there are some similarities. CETP inhibition is considered in the setting of sepsis ( PMID: 33228395 • DOI: 10.1161/CIRCULATIONAHA.120.048568). As explained by the authors of this review, many more aspects of virus biology are influenced by HDL. The question is whether HDL is having an effect on the course of COVID-19 disease in selected individuals and whether this hypothesis can be tested by an intervention study.

Author Response

Point-by-point answers to reviewers

We thank the reviewers for their positive comments about our manuscript and for their helpful and constructive critiques. We enclose, below, a point-by-point response to each of the comments raised by the reviewers:

Reviewer 2

  1. What is now the official correct nomenclature of Scavenger receptor class B type I (SR-BI) at protein level? It used to be SR-BI and not SR-B1. Did it change? To my knowledge, Monty Krieger insisted that it was SR-BI.

SR-B1 is the official correct nomenclature of the Scavenger receptor class B type I at the protein level, as laid out in the following papers: Prabhudas, et al. (2014) Standardizing scavenger receptor nomenclature J. Immunol. 192:1997-2006. Doi: 10.4049/jimmunol.1490003; Prabhudas et al (2017) A consensus definitive classification of scavenger receptors and their roles in health and disease. J. Immunol. 198:3775-3789. Doi: 10.4049/jimmunol.1700373, both of which are co-authored by Monty Krieger.  

  1. Systemic inflammation, as measured by C-reactive protein, is a predictor of venous thromboembolism, acute kidney injury, critical illness, and mortality in COVID-19. Therefore, it is not surprising that low HDL cholesterol is a univariable predictor of venous thromboembolism, acute kidney injury, critical illness, and mortality in COVID-19 or in other words of the severity of COVID-19. Furthermore, many subjects with severe COVID-19 have multiple co-morbidities that result in low HDL cholesterol. Low HDL cholesterol may an epiphenomenon (when there is no relation with outcomes) or there may be confounding factors (when there is a relationship with outcomes). The in vitro data and cell biological data are consistent with a potential causal role of HDL, but this does not prove in vivo causality.

We thank the reviewer for pointing out this important caveat. We fully agree with the reviewer and have more clearly stated this caveat in the text of our review.

Page 3, Section 1, lines 61-63: “The above observations are consistent with HDL exerting a protective effect in COVID-19 disease, although the alternative, that low HDL may be an epiphenomenon, cannot be ruled out.”

Page 24, Conclusions section, lines 627-631: “It is important to point out, however, that while associations of HDL levels with COVID-19 disease severity as well as in vitro cell biological data may be consistent with a potential for HDL to influence SARS-CoV-2 mediated disease, they do not prove in vivo causality. This awaits in vivo pre-clinical and, if warranted, clinical intervention studies.” 

  1. If one considers HDL particle concentration as a determinant, one should also consider that the biological response will be dependent on the HDL proteome and lipidome (and mIRNA’s in HDL).

The authors agree with the reviewer, and we have made efforts throughout the manuscript to emphasize this concept.

  1. All non-standard abbreviations should be explained at first use (e.g. ABCA1, ABCG1,…). These are known by experts in the HDL field but maybe not by other investigators and clinicians dealing with COVID-19.

We apologize for this oversight. We have now explained all non-standard abbreviations on first use throughout the manuscript.

  1. COVID-19 is clearly distinct from sepsis but there are some similarities. CETP inhibition is considered in the setting of sepsis (PMID: 33228395 • DOI: 10.1161/CIRCULATIONAHA.120.048568). As explained by the authors of this review, many more aspects of virus biology are influenced by HDL. The question is whether HDL is having an effect on the course of COVID-19 disease in selected individuals and whether this hypothesis can be tested by an intervention study.

The authors thank the reviewer for bringing up this excellent point and for highlighting the study of CETP inhibition in the context of sepsis, which we have now incorporated into our discussion as well as the concept that such an intervention study in the setting of COVID-19 would be an important test of whether HDL may affect the course of disease in this setting.

Pages 17-18, Section 6, lines 412-421: “Similarly, in a mouse model engineered to express CETP, pharmacological inhibition of CETP, leading to increased HDL-cholesterol levels, suppressed sepsis induced inflammation and improved survival.  In agreement with this, CETP gain of function mutations in humans were associated with increased mortality due to sepsis, while reduced CETP function was associated with increased survival.  CETP exchanges cholesterol ester on HDL for triglycerides from other lipoproteins (Figure 2); therefore, CETP inhibition or reduced CETP activity results in increased HDL cholesterol levels, whereas increased CETP function is associated with lower HDL cholesterol. This demonstrates, at least in the setting of sepsis, that HDL-targeted therapy may have beneficial outcomes.

Page 23, Section 8, lines 542-547: “For example, if the reductions in HDL levels which occur during the course of COVID-19 disease impact on the severity of disease, then interventions aimed at raising HDL levels may be beneficial. In this light the recent reports that CETP inhibition is beneficial in sepsis, at least in mouse models, suggests similar interventions may have beneficial impacts on COVID-19 disease.”

Page 24, Conclusions section, lines 588-592: “It is important to point out, however, that while associations of HDL levels with COVID-19 disease severity as well as in vitro cell biological data may be consistent with a potential for HDL to influence SARS-CoV-2 mediated disease, they do not prove in vivo causality. This awaits in vivo pre-clinical and, if warranted, clinical intervention studies.” 
